# Scaling Digital Health Innovation: Developing a New ‘Service Readiness Level’ Framework of Evidence

**DOI:** 10.3390/ijerph182312575

**Published:** 2021-11-29

**Authors:** Janette Hughes, Marilyn Lennon, Robert J. Rogerson, George Crooks

**Affiliations:** 1Digital Health & Care Innovation Centre, University of Strathclyde, Inovo Building, 121 George Street, Glasgow G1 1RD, UK; george.crooks@dhi-scotland.com; 2Department of Computing & Information Science, Livingstone Tower, University of Strathclyde, Glasgow G1 1XQ, UK; marilyn.lennon@strath.ac.uk; 3Technology & Innovation Centre, Institute for Future Cities, University of Strathclyde, 99 George Street, Glasgow G1 1RD, UK; r.j.rogerson@strath.ac.uk

**Keywords:** service innovation, service readiness, scaling, digital health innovation, evidence, evaluation methods, case for scale

## Abstract

Digital innovation has scaled exponentially in many sectors including tourism, banking, and retail. It is well cited that the health sector is slower to embrace digital health innovations (DHI) beyond the pilot stage and consequently, many successful DHI pilot projects have failed to scale up. Such failure arises in part from a knowledge gap around what type and level of evidence are needed to convince implementers and decision makers to fund, endorse, or adopt new innovations into care delivery systems and sustainable practice. Much is known about the range of DHI evaluation methods used; however, less is published on the evidence that decision makers need to move innovations to scale. This paper draws on interviews (N = 18) with decision makers/project leads engaged in DHI in Scotland to identify what evidence matters when making DHI adoption/scale decisions. The results are used to present a heuristic service readiness level (SRL) framework that captures the changing nature of the evidence base required over a project lifecycle for progression to scale. We utilise this framework to discuss ‘what evidence’ is required and ‘how data accumulate’ over time to assist project teams to build a ‘DHI case for scale’.

## 1. Introduction

In the last 20 years or so, various industry sectors have transformed themselves by capitalising on new and innovative digital technologies, creating unprecedented benefits, efficiencies, and convenience advances to common services such as banking, tourism and retail [1]. Adopting such technological innovation has not only demonstrated improved efficiencies in service delivery and enhanced operational performance but has also led to increases in convenience and customer satisfaction and generated behavioural change. In contrast, the uptake of digital health innovations (‘DHI’, also referred to as Telehealth, Telemedicine, Mhealth, Digital Health) has been much slower, more incremental, and less demonstrably scaled (spread to other/more contexts and/or wider populations/numbers of people). This led Standing et al. [2] to conclude that the scale and scope of telehealth have not become ‘business as usual’ in the health sector.

In this paper, we explore one dimension of the process of scaling up DHI, focusing on the nature of the evidence base that is required by key decision makers to translate pilot schemes into wider adoption and cases for scale. Through interviews with decision makers and project leads, we mapped this evidence to DHI examples and associated ‘levels of readiness’ for digital health innovation. In doing so, we have produced a heuristic service readiness level (SRL) framework that can be used by project teams and commissioners to ensure all parties gather the right assurances and evidence throughout the process to scale DHI if and when appropriate. Following a review of previous research on scaling digital health innovations and an outline of the methods used, this paper focuses on the different themes and evidence bases which are required to support scaling. Together, this allows us to present a new framework, based around service-level readiness, which offers insights to the varying and desirable evidence sought to assist in building a case for scaling DHI. This paper concludes with a brief discussion of the implications and potential use of the framework.

## 2. Related Works

Scaling DHI is not a new issue and has been discussed in many publications over the last ten years. The related work is therefore described in the following section as part of the literature review relative to the key benefits, challenges, and approaches to scaling up DHI.

### 2.1. The Benefits of Scaling DHI

The benefits of adopting DHI have been widely acknowledged, offering opportunities to be ‘transformative’ [3] and even ‘disruptive’ [4], reshaping organisational activity and strategies as well as altering consumer behaviour. In March 2020, the Healthcare Information and Management Systems Society (HIMSS) concluded that adopting DHI can transform delivery, stating that ‘digital health connects and empowers people and populations to manage health and wellness, augmented by accessible and supportive provider teams working within flexible, integrated, interoperable, and digitally-enabled care environments that strategically leverage digital tools, technologies and services to transform care delivery’ [5] p. 24.

It is accepted that DHI can play a significant role in modernising health care and that digital health technologies (at least in theory) lend themselves to scaling to the benefit of the populace [6]. For at least the last decade, however, there has been industry wide concern that digital health innovation continues to be constrained to smaller-scale, often local project pilots. More work is therefore needed to identify what factors might facilitate the adoption and scaling of promising new digital health innovations. Significant change within the digital health sector is also necessary in order to gain the commercial advantage that comes with the transformation of innovations at scale for public-sector services.

### 2.2. Challenges in Scaling of DHI

The adoption of technologies for health and care in general has been far from a smooth process, with innovation and transformation using digital for this sector often viewed as a ‘huge challenge’ [7]. To date, digital health innovation has been predominantly experienced through the prism of small, local pilot studies which have demonstrated to some level the potential for greater adoption but have failed to achieve scale. Kuipers et al. [8] note that globally there has been a reliance on pilot DHI projects as a means of informing policy and service innovation and an accompanying disappointment that even when successful, they have not been capable of bringing sustained change to the broader service provision. Such affliction of ‘pilotitis’ [9] continues to characterize this sector globally and is noted frequently as a key frustration for those involved in the sector.

In explaining this absence of scaling in the UK, Lennon et al. [10] suggest that there is a need for greater investment in national and local infrastructure, implementation of guidelines for the safe and transparent use and assessment of digital health, incentivisation of interoperability, and investment in upskilling of professionals and the public. Labrique et al. [11] suggest that scaling up of DHI can be enhanced where there is a clear need to generate tangible benefits, and engagement from all stakeholders from early stages, the provision of infrastructure to support DHI, and where initiatives need to be simple and adaptable. These authors also suggest that scaling is more likely to occur where there is alignment with broader health policy, underlining other studies which point to limitations existing at a policy level. Desveaux et al. [12], in their review of Canadian DHI, suggest that at a policy level, barriers to upscaling include the absence of a system-level definition of innovation to enable alignment of innovation with service delivery goals across health organisations. They also argue that there needs to be more clearly defined organisational roles, in order to champion innovation adoption and for improvements in the coordination of various types of knowledge within health and policy systems.

In the context of DHI, complexity is not only centred on organisational and operational contexts, but also on the nature and form of evaluation required to reflect the wide array of health-related settings covered by DHI. A European Commission report [13] noted, ‘given the diverse forms, usages and impacts of digital technologies in health care (ranging from general use of computers to algorithms designed to assist radiologists and radiotherapists in detecting and treating cancers, from robotic surgery to computer aided decision models, and from mobile device apps helping patients to self-manage their disease to electronic health records), this requires evaluations on different levels’ [14] and in turn different sorts of evidence.

### 2.3. Strong Evidence Base

This inherent complexity can be problematic for the project team to collect the correct evidence that data decision makers require. In an attempt to address the issue of complexity in the UK, the Medical Research Council (‘MRC’) established a framework for complex evaluations in 2008, which is due to be updated later in 2021. Although presented as an iterative approach, this framework emphasizes four phases of a project lifecycle: intervention development; feasibility and piloting; evaluation; and implementation. Notwithstanding the merits of this framework and the insight that assisted this study, it is unclear to project teams what evaluation methods should be adopted at a particular stage, and what is the expected level of evidence and assurance required in relation to how ready the service is to adopt the DHI; what we term service readiness levels (SRL, see Appendix A). Furthermore, it is difficult to ascertain the exit assurance criteria for decision makers and/or senior responsible officers, in order to facilitate the move to the next level, i.e., towards adoption and scale within routine service delivery.

Decision makers require to understand whether a DHI is viable and capable of being adopted and scaled. To focus some of the uncertainties and interdependencies which accompany innovation adoption, researchers have developed the Non-Adoption, Abandonment, Scale-up, Spread, Sustainability (‘NASSS’) Framework [15]. This framework is designed to support the integration of theoretical perspectives, technology adoption, organisation change, and system change to build a framework that helps predict and evaluate successful upscaling of DHI. Empirical application of NASSS has underlined its value in enhancing an understanding of failure to adopt or spread, emphasising that complexity associated with DHI is a strong inhibitor to widespread adoption [16,17,18] as well as the difficulties in generating an accumulating knowledge base for guiding decisions about DHI [19].

In pursuing the efficacy of scaling a DHI, there remain issues around how research and empirical evidence is generated through project lifecycles. The lack of understanding on how the evidence is aggregated, iterated and built upon to create a final case for scale can be daunting to inexperienced DHI project staff. Projects should plan from the early stages to not fall into the “no evidence, no implementation—no implementation, no evidence” paradox often seen in digital health and highlighted recently by Guo [20] as a key challenge that needs to be considered to help project teams move this forward.

In this paper, the questions we seek to address include what is the nature of the evidence base required across the entire project lifecycle to enable decision makers to be confident when considering scaling up interventions? What are the stages of development at which different decision makers are involved in evaluation and engaging with evidence bases? How can key decision makers become the ‘champions’ that Desveaux et al. [12] suggest can coordinate across whole systems? In addressing these, and as a contribution to the global drive to scale DHI, this paper summarises the key findings from a study that led to the development of a heuristic DHI service readiness level (SRL) framework. This provides an easy way to detail the evidence required at each stage of the decision-making process, how evidence accumulates as projects mature, that ultimately assists in building a robust evidence base that supports a case for scaling DHI.

## 3. Aims and Methods

The aim of the research was to explore, identify, and understand existing and appropriate future evaluation methods and evidence for DHI, in relation to building cases for scale. The end goal was to establish a usable framework which corresponds with the readiness levels identified through the common DHI project lifecycle stages and evidence required to be gathered at each stage.

In order to achieve our objective of establishing a reliable and usable framework, the authors adopted a qualitative approach in the form of (i) a scoping review of the literature using the methodology set out by Arksey and O’Malley [21], (ii) semi-structured interviews (using the ‘Kvale’ method along with conceptual guidance from Rowley [22]) with N = 18 mainly Scottish DHI key stakeholders, project leads and decision makers. This allowed us to unearth and consolidate insights not readily available in existing publications and produce (iii) mapping of the interview themes and evidence quoted to the different levels of service readiness to create a SRL framework.

### 3.1. Interview Method

Overview/briefing paper detailing the purpose of the research, emailed in advance to interviewees (N = 18). The interviewees were drawn from senior management involved in DHI in Scotland, covering finance (2), clinical care (9), service management (3), and technical (4) (interviewees are anonymized using coding to represents their key skill; F—finance, C—clinical, S—service and T—technical). Their selection covered the main institutions involved in DHI decision making, including Scottish government, NHS, Innovation group project teams. In relation to the length of DHI experience of those interviewed, 15 interviewees had over 10 years’ experience in this field, 2 had over 5 years and 1 had under 5 years, with a gender split of 50% male and 50% females. The interviews were semi-structured, encompassing their experience with DHI projects, examples of good practice, barriers to scaling and their views of how scaling might be advanced (see Appendix B for interview questions) and took place over a 24 month period (from 2018 to 2020).Digital audio recordings of the interviews for transcription and later thematic analysis and coding.Field notes undertaken during the sessions, with key words and themes highlighted.Reflective thematic analysis—deductive [23], which followed the process of data familiarisation, data coding to generate key themes, supported by NVivo (V12).

### 3.2. Framework Development Method

Interview content mapped and linked to different stages in the DHI project lifecycle—identified through the illustrations offered by those interviewed as part of this study.Framework constructed in line with the existing NASA technology readiness levels (TRL) [24] framework in using the same analogy and principles.Service readiness levels (SRL) described using specific headline titles that were summarised from the illustrations offered and leading examples of DHI projects that had or were moving towards national scale. These SRL titles were then arranged in chronological order and titled to describe the types of activities and journey observed, as per the initial interview content. The evolution of this framework took place with title headings changing and reordered where necessary as part of the latter consultations and feedback when the SRL framework (see Appendix A) was being initially validated.This SRL framework was then tested and validated by N = 14 interviews with key DHI leaders to gain further feedback and detail to optimise usefulness of the framework. These interviewees included 5 who were involved in the original research as well as an additional 9 who had comparable senior management roles in relation to DHI in Scotland, and these interviews took place over 2020/21.

## 4. Findings

The research highlights two key elements to DHI scaling. First, it was evident that there were a number of thematic areas where a strong evidence base needed to be developed to provide the confidence and robustness required to support upscaling. However, second, it was also clear that this of itself was not sufficient to enable scaling, as the evidence had to be aligned to different stages of project development. This was strongly related to the notion of service readiness. In this section, we explore further these two strands.

### 4.1. Thematic Evidence Bases

Analysis of the interviews identified six main evidence-related priority themes that were regarded as important by decision makers in reviewing a DHI case for scale (Table 1). Together, they provided the evidence base which they felt would offer assurance that the DHI was a good investment, could be adopted safely, and was considered to be ready for implementation at a national scale. Each encompassed a number of sub themes that reflected varying dimensions of the evidence base required amongst the interviewees and the barriers to scaling that evidence had to overcome. The following section exemplifies the nature of the evidence considered important, drawing on comments from those interviewed. The participants are identified by codes to retain their anonymity. This section also reveals how the value and content of such evidence is dependent on the stages of readiness within an organisation or decision-making process—a point we elaborate on in the subsequent discussion section.

#### 4.1.1. Service and Organisational Evidence

In terms of organisational capacity to move to scale, the interviews revealed that this stage was often unclear, firstly on which overarching organisation would take ownership for building the case for the national scale and secondly who would be ultimately responsible for making the decision to scale nationally. It was noted that NHS Boards in Scotland did not think they had the mandate, capacity, or capability for building a national case for scale. This was said to cause delays and diminish pace, momentum, and morale, especially when project teams and partners believed that they had ‘good enough’ evidence for scale at that point. As one key stakeholder noted:


*“there are very few people who could really change business aspects of the service…. And then there’s even fewer people who are given the authority and the capacity to actually take action. So even the people who are interested in the change and can express what the future should be like, don’t really have the means to move forward to scale” (T&X2).*


In making a case for scaling therefore in terms of organisation and service dimensions, evidence at the initial project stage in understanding demand and conveying a strong vision was particularly significant. This helped address questions as to whether a DHI would make a difference to the service, and create enough interest nationally against a major strategic priority, if scale was the end goal. As *C&Y2* expressed it, the evidence needed to answer


*“does it solve or contribute to our real challenge in the system…so people see its value?”*


Or as C&Y4 noted, it provided assurances that the DHI *“addresses demand and capacity challenges”*.

Evidence on the benefits of introducing a new transformed service using DHI were deemed essential. Each interview raised the need to articulate benefits from different perspectives, including benefits to (1) them, at a personal level, (2) a service level and (3) for their customers, ‘patients’ and ‘carers’. It was accepted that a range of common benefits would be expected, with hospital and health demand metrics rating most highly in persuading decision makers of the value to scale.

Alongside this was a desire for evidence to reassure service delivery would not be compromised by introducing DHI, demonstrating that service quality should be at least maintained, if not improved, using DHI. To this end, there was broad agreement that clear baseline evidence of the current service provision and performance metrics from multiple viewpoints was desirable in any case for scale, allowing opportunities for transformation options to surface early in the process, and to enable comparisons to be made between traditional approaches, pilot studies and future states when delivered at scale.

Beyond this, concern was raised about service continuity for projects aiming to reach scale fast, and as to how the project would be sustained while it awaited review and subsequent approval for national case for scale. Interviewees noted a frequent lag between project pilot conclusion and moving to a business-as-usual (BAU) service; the latter requiring specific skills, agreements, and a level of support for continuity to be confirmed to ensure the organisation responsible was confident to offer this as a service. As C&X1 expressed it


*“you have to draw on quite a lot of know-how, I think you’ve got to have quite a range of skills to actually bring that together and to put it into something as business as usual”,*


Creating an inherent reluctance by organisations to decide on scaling up.

The complex inter-organisational working that typifies decision making in the health sector means that additional evidence is needed to assist multi-stakeholders in the co-designing process, including agreement over the desired scaled up form of the DHI. Evidence mapping that demonstrated a review of best practise—referred to as ‘landscape/literature reviews’ and ‘horizon scanning’—from other stakeholders, regions, and countries was noted as imperative. Such evidence at different stages helped to ensure a strong reference base, allowed shared learning, and offered reassurance practices were ‘not reinventing the wheel’.

Interviewees stated it was easier to convince those that have the authority to ‘take action’ if there was real-world evidence (RWE) on the benefits/impacts and also if there was guidance on ‘how to do it’. Strong proof that it was achievable, and the service would accept a ‘new way of working’, was also important to build confidence with the decision makers. There was a general sense that evidence of this kind, often a process evaluation and softer related evidence (views and endorsement), was often either missed or lacked depth and clear sponsorship, resulting in decision delays. As projects moved towards implementation at scale, there was a need for practical evidence to allow for clear guidance on ‘how to implement’ the DHI and ‘create the conditions’ for the service change to be understood, realised and endorsed internally.

#### 4.1.2. Clinical Evidence

Further, everyone interviewed recognised that clinical efficiency was a top priority for reviewing a case for scale, and that fundamental pieces of evidence on clinical efficacy and patient safety were key, including an evidence base that enabled opportunities to:


*“re-evaluate its position around what conditions can we make to make sure that it is safe, and that it is complying” (S&X1).*


Predominantly, the interviews revealed that there was a strong preference that as the DHI project matured to a higher state of readiness that this evidence was in the form of clinical trials; seen as generating a strong and robust clinical evidence base that would include efficiency and effectiveness and would withstand clinical scrutiny. The clinical trials discussion featured in many of the interviews with much debate on the best method and whether ‘Randomised control trials’ (RCTs) referenced as the ‘gold standard’ was appropriate for DHI. It was noted by most that a hybrid of a pragmatic RCT was best used, if the service criticality required it as evidence, noting not all DHI require this level of evidence if the risk of harm to the patient is considered to be low.

There was also consensus that evidence of other clinical benefits was both desirable and often necessary to achieve support and reduce the barriers for scaling, this often manifested itself in benefit plans and cost benefit analysis formats. The demonstration of clinical effectiveness and the delivery of better use of resources were essential. Such evidence was important if the necessary clinical leadership were to be achieved:
*“…without this clinical backing it unlikely that even if an innovation is proved to be effective, efficient and convenient for end users it will be too difficult to mandate without clinical leadership and champions being in place to promote to their peers and drive this forward as an acceptable option” (T&X1).*


Such leadership not only had to endorse the adoption and scaling DHI, but to feel a degree of ‘investment’ and ‘ownership’ in the opportunity.

#### 4.1.3. Finance, Legal and Standards Evidence

Unsurprisingly, financial issues along with legal and standards evidence are viewed as key indicators when introducing a DHI. Interviewees stated that clear evidence of the cost of the new service was a vital piece of data required for the case for scale, closely tied with the return the DHI would provide in respect to the investment. Interviewees highlighted a range of cost benefits that could be evidenced were discussed, including ‘cost savings’, ‘cost avoidance’, and ‘cost neutral’; all of which would potentially lead over time to a cost benefit. In assessing such benefits, evidence that demonstrated value through a more holistic approach and ‘whole system’ view was seen as advantageous:
*“...you can’t look at these projects just as health projects in isolation, they have an economic dimension, education and industry dimensions, various facets to them that need to be taken into account”. (F&Y1).*


A major sub-theme raised under finance evidence was in connection to affordability to launch the new service and the sustainability of scaling, this related to the service and costs associated in moving it into a business-as-usual (BAU) state, which in most cases was noted to require a recurring budget. The noted BAU challenge and often barrier identified were linked to references of short-term funding structures used for projects of this nature. The realisation being that immediate savings would not be achieved in the short term with systemic political and organisation constraints. Common barriers included, budgets unable to be redirected easily or quickly and due to the lack of dedicated parallel innovation budgets for DHI sustainability being in place within most governments and service organisations. This was exacerbated by the need to evidence compliance, relating to legal, procurement, regulatory (e.g., medical device directive) and general sector specific standards.

Whilst robust evidence of cost benefits and cost effectiveness would be a strong driver for DHI scaling, the interviews revealed that opportunity cost pressures and other challenges around risk and liability meant that there was a reluctance to commit to scaling without clearer evidence of future implications, with a gap identified within the DHI case for scale being the ‘consequence of not scaling’. To date, little research exists about the future implications of not approving a case for scale with regard to DHI.

#### 4.1.4. Citizen Evidence

Although clinical and medical evidence formed the basis of much of the evidence base sought for scaling, there was an appreciation that evidence of support from other stakeholders, citizens and political decision makers should form part of an evidence framework.

In making the case for citizen support, it was evident that this label was commonly applied to describe not only patients, but also health care customer (including carers), and service users, although for some interviewees this was broadened to the wider population. There was general agreement that the DHI should be evidenced to be ‘patient centric’, regarded as a crucial principle that strongly aligns with government strategies for this sector. Clear evidence that patients had been consulted and were part of the co-design for the potential future service was made clear by those interviewed, along with evidence on usability, acceptability, along with uptake/adoption and demonstration of equitable access to prove citizen approval. In short, as C&X3 expressed it: *“I’m looking at evidence of workability, acceptability”*.

That said, across the interviews, there was also a desire for evidence that assessed demand and quality of a new DHI enabled service from a citizen’s perspective. For most, this indicated not only likely uptake of the service but also linked with the health services’ agenda of empowerment. The type of evidence of citizen benefits sought included real stories of the difference this had made to patients and citizens, or the benefits to others such as ‘carers’ lives, which also highlighted how their quality of life was or would be improved by this new service innovation. Importantly, interviewees also noted that if the DHI concept was found to be unpopular with the public, then organisations and governments would be unlikely to support the DHI for scale.

#### 4.1.5. Political Evidence

The political dimensions of introducing a DHI for the national scale mattered. With most DHI cases for the national scale in Scotland having to be reviewed and funded, at least at the initial scaling stage, by Scottish government. There was a compelling case for evidence which showed alignment with government strategy and policy priorities, with clear critical success factors and vision required to be set out early as part of the case for scale. Further, it was desirable that project teams should ensure that there was clear and easy to digest evidence to communicate alignment with policy and preferably a confirmed senior sponsor within government, especially as the state of readiness was maturing—what one respondent termed *“political enthusiasm” (F&Y1)*—to provide endorsement and backing to evidence how this aligns with strategic priorities and major challenges in the case for scale was seen as essential.

#### 4.1.6. Technology Evidence

The final theme—and one that underpins all DHI initiatives—relates to the technology that enables the introduction of a DHI service. Surprisingly, technology evidence was mentioned least by the interviewees; suggesting that evidence of the existence of technology and support was being ‘taken for granted’.

Those interviewed made it clear that the level of technology evidence was dependent on where the innovation was in relation to the technology readiness level (TRL). Existing technologies that had a proven track record and a strong evidence base required less specific technological evidence than a technology that was emerging and was regarded a disrupter, the latter often lacked a robust evidence base within this sector, regarded as ‘not trusted’, leading to heightened risk when considering a case for scale. The technology had to be proven with evidence that it was safe, accessible, usable, and reliable. Further, it was desirable for technical endorsement to be demonstrated for all users: citizens (patient and carers), professional staff (clinical, admin and service management) and the NHS board health IT departments (Ehealth).

Interviewees flagged a necessity that for to be both interoperable and able to be integrated easily into existing NHS legacy systems if required:
*“in terms of evidence … you’re buying a thing that isn’t connected to other things. There’s the whole interoperability.... You’re having to plug something into our existing infrastructure” (T&Y1).*


However, it was equally important that there was clear evidence that the technology could flex with local needs, proving adaptability and transferability to other regions and conditions in building a case for scale would be beneficial. This is turn offered a degree of what was termed as ‘future proofing’ and contributed to the ‘economies of scale’ argument within a ‘case for scale’.

### 4.2. Project Lifecycle Evidence

What emerged from the interviews and is currently absent from previous research is an appreciation that the nature of evidence varies dependent on the stage in the lifecycle of the project, criticality of the service, and state of readiness to accept and adopt and scale the innovation. Cross-cutting the thematic evidence bases noted above in Section 4.1, the interviewees highlighted different points in their decision making or development of DHI evaluation at which they drew upon an evidence base at particular stages, this was highlighted in the interviews:
*“there’s lots of different people involved at different stages” and that “the business case was only the end point of quite a long process” that “targeted multiple different structures, each with a different purpose … continual throughout. So, it’s not a position where you’re delivering a business case, which then is a surprise to people. Actually, the reality of the business case was almost decided” (C&Y5) due to decisions being made along the way with “a lot of people to convince” (C&Y2) to accumulate a sense of “confidence and assurance” (S&Y1).*


In our analysis, we identified that there was an emerging view that the form and nature of evidence required could be conceived of as being linked to different degrees of service readiness levels. DHI service readiness levels (SRLs) is a framework that supports the assessment of the maturity of an organisation in adopting a digital health innovation into a health service. The SRL approach and format was constructed from that of the NASA TRL [24] structure and key principles which was originally developed in the 1970s. Further details of the specific level descriptors can be found in Appendix A.

Using this general SRL model and insights from the interviews, we have created a heuristic framework which details the varying nature of evidence that should be gathered at each stage of the DHI project lifecycle, corresponding to the different levels of SRL. This provides a simple representation of how best to collect the evidence, demonstrating that the evidence accumulates as the project matures to ultimately build a case for scaling the DHI.

In constructing this framework (Table 2), we have drawn on the wider notion of different service readiness stages in line with the DHI project lifecycle that follows the concept from an early idea to the DHI being implemented at scale within a service (Column 1). This allowed the findings as per interviews (evidence themes) to be mapped to the different forms of evaluation methods (Column 2), evidence and assurances required (Columns 3), as well as the criteria which would enable a confident exit criteria (Column 4), to a higher readiness level (if deemed appropriate).

The titles and contents of these columns reflect the thematic evidence findings with key stakeholders as part of the interviews conducted and were synthesized and organized to allow the researchers to present the findings in an easily to understand format. This provided a clear ‘staged approach’ to build an evidence base that would enable progression if the criteria was met, to exit points (as per column 4), allowing diligent movement towards the next stage of readiness and eventually to scaling the DHI (if appropriate), please note it was found certain stages can be iterative (SRL 6 and 7) and some may run in parallel (SRL 3 and 4) and other stages required to be refreshed (SRL3) at the final case for scale (SRL 8), dependent on the time lag between (SRL3 and SRL8).

## 5. Discussion

The development of the above framework offers a novel and easy way to detail the evidence required at each stage of the decision-making process, how evidence accumulates as DHI projects mature, that ultimately assists in building a robust evidence base that supports a case for scaling DHI. It also provides insights into how those involved in DHI, as Desveaux et al. [12] argue, are desirable, can help to champion project extension through development of appropriate, robust evidence, accepting that such bases are shaped by different stages of readiness.

It is acknowledged that further research and application of the framework are clearly needed to test and validate further such assertions but given the current difficulties in scaling up small DHI pilot studies, we suggest that this framework provides a significant contribution to enable an appropriate robust evidence base to be generated to assist scaling. However, one implication of bringing together the different thematic forms of evidence required to make robust cases for scaling with different stages of project development and service readiness is that it is feasible to envisage an overarching framework of evidence for DHI scaling. Table 3 offers an initial example of such a combination framework. This shows how different evidence themes (first column—in blue) can be aligned with the service readiness level stages (columns labelled 1 to 9 in gold), providing a neat summary of the evidence that has to be gathered at each stage, and accumulated over the project lifecycle to ensure enough of an evidence base is present as part of the ‘case for scale’ for decision makers to review and approve if applicable for scale. Such a framework has the opportunity to help steer the development and evaluation of smaller-scale and evolution of pilot projects, which dominate DHI presently, to become more mainstream and larger scale.

## 6. Conclusions

In contrast to the successful transformation of different industry sectors through the adoption of new and innovative digital technologies, the arena of DHI has been marked by small-scale pilot projects unable to be scaled up to create health and well-being benefits and service efficiencies. The existence of a mature technical solution does not guarantee that such a product will be adopted by a health care system and certainly does not guarantee widespread adoption into business-as-usual service delivery and or scaling across a whole institution or national care delivery system. This paper has sought to fill a major knowledge gap around what type and level of evidence are needed to convince implementers and decision makers to reduce the barriers and advance large-scale DHI.

Technical readiness levels (TRLs) have become accepted internationally as way to classify the evolution of a technology since proposed by NASA in 1973. We have highlighted that of equal, if not greater importance is a systems willingness and readiness to change and adopt a DHI. To secure a digitally supported and enabled future, we must support organisations to better understand where they are starting from on their journey, how ready they are and what evidence they need to gather to secure successful adoption and scaling of DHI.

Through interviews with key stakeholders, we have created and tested the potential utility of a heuristic service readiness level framework to detail the type and range of evidence sought to allow DHI scaling to be more easily assessed. This offers for the first-time clear pathways in how evidence can be accumulated to enable scaling up of digital health innovation and building a robust ‘case for scale’ for decision makers. Further research on the application of the framework is needed to test this SRL framework and understand in more detail if there are gaps that could be further progressed to evolve this study and aid the progression of scaling DHI.

## Figures and Tables

**Table 1 ijerph-18-12575-t001:** Evidence priority themes to support DHI scaling.

Themes	Sub-Themes
Service/Organisational (400 references)	Service demand and vision Service quality Current service understanding Future preferred service transformation Service benefits and impacts expected Service change, implementation, and transferability
Clinical (300 references)	Clinical acceptance Clinical effectiveness and better use of resources Clinical efficacy and patient safety Leadership and ownership
Finance, legal and standards (183 references)	Cost and return on investment Value for money, including procurement approaches Affordability and sustainability Risk, benefits, liability, and standards/regulations
Citizen (95 references)	Citizen experience Citizen demand and empowerment Citizen benefits
Political and policy (91 references)	Strategy alignment Political guidance and sponsorship
Technology (68 references)	Existing and disruptive technology Acceptability, usability and accessibility Interoperability, adaptability and integration

Source: authors’ interviews.

**Table 2 ijerph-18-12575-t002:** Service Readiness Level framework aligned to Evaluation, evidence summary and assurance/exit criteria (* multiple iterations and cycles may be necessary).

	Service Readiness Levels (SRL)	Evaluation Methods	Evidence Summary	Assurance—Exit Criteria
	SR9—Service change implemented	Normal service change control process and evaluation methods should be followed	The service is implemented into Business as Usual and will follow normal evaluation and improvement practice for refinements, support packs in place.	New service accepted as BAU—business continuity/improvement/SLAs in place.
	SR8—Case for scale	Parallel run required between the project team and the Service implementation/change/business as usual team	The Service/BAU team must feel comfortable with the evidence before the service is onboarded in a live environment and offered at scale.	Case for Scale—Sign off by implementing organisation and national funder (often Government)
		Process, Clinical, Economic, financial and technical evaluation substantiated with qualitative feedback from clinicians, service manager, Ehealth, finance/legal/policy execs and customers (Citizens—patients/carers, popn).	Process, finance, economic evaluation evidence including technical due diligence evidence. Implementation/Set up Pack, Blueprint and sustainability plan. Benefits realisation/impact case—as per CSF. Business continuity plan. A full business case could be built, or further proposals to allow the innovation to be transferred for more testing/iterations *—to test transferability.	Sign off by programme board SRO—to progress for national scale commitment. The SRO must be assured that all evidence is present, endorsed by boards generally that it is regarded a sound case for investment.
**Multiple iterations ***	SR7—Evaluation and Evidence gathered	Process, Clinical control trial (RCT variations—pragmatic), CBA/ROI/Cost effectiveness/Cost consequence/Cost utility, Economic impact analysis, HTA, Surveys/interviews (Users, Clinical, service etc.) PROMS/PREMS, QALY, Comparative and consequential studies, QoL, HRQoL, EQ-5D (EuroQol—5 Dimension), Carbon footprint analysis.	Report findings on effectiveness, safety, acceptability, affordability and sustainability, comparators from current state to new service state, comparators with other regions. Test for Change report Patient data on experience and outcomes. Quality of Life, Quality of service, specific metrics driven related to outcomes and impact e.g., reduced—waiting times, bed days, falls, exasperations, Net zero—carbon emissions etc.	Sign off by project team and programme board. The SRO must be content that the evidence is sufficient to allow either for the full business case, or a subsequent proposal that evolves the DHI for further adoption testing with other health boards.
	SR6—Real World Evidence testing	Basic service, economic and financial modeling—CSF made clear. Service Simulations and blueprint/process evaluations methods considered.	Small pilots (case for testing articulated)—aggregating previous info and presenting current RWE findings. Simulation can be used at this point. Test for change (TEC) activated if required, CSF must be clear at this point.	Sign off by project team and programme board, SRO commitment demonstrated to invest resources with a pilot.
	SR5—Future state accepted in principle	Usability/Accessibility testing/EQIA, Acceptability testing, Interviews, and surveys, Future mapping methods, Net zero contribution analysis.	High level Evidence gathered that it is/and will be generally accepted within work practices, can be used effectively with ease, is intuitive and does not cause extra work and importantly create benefits. Endorsed by a range of stakeholders (Org, Clinical, patients/citizens, political, finance/legal/standards including procurement approaches and technical aspects).	Sign off and assurance from professionals—clinical, EHealth and service staff as an acceptable future option that warrants RW testing—weighted against levels of risk/opportunity/benefit to the system.
**Parallel and iterative**	SR4—Future state (FS) options co-designed	Simulation, paper-prototype, participatory co-design workshops/insights—persona/storytelling methods and visual illustrations.	Service redesign options and digital opportunities explored, pathway reviews and opportunity options appraised. High level FS blueprint drafts. Case studies/storytelling/personas used to communicate the future state options with possibilities linked to infrastructure/interoperability implications	Sign off at professional level that the FS options have been validated, supported by patient views/feedback—senior sponsor endorsement and assurance is in place.
	SR3—Horizon scanning	Landscape/literature/market review—Market analysis; best practice, Desk research, rapid review, Interviews/Surveys, Champions	Publications/Reports on similar services and innovations—horizon scanning. Competitive analysis—past evaluation/evidence data of innovation—used, tested, implemented. Empirical evidence gathered is appropriate (systematic reviews referenced or conducted). Art of the possible articulated.	Sign off by project team that desk research best practise has been reviewed and there is assurance that an appetite at snr. Level in the organisation/system to promote change (e.g. new working).
	SR2—Current state (CS) understood/accepted/validated	Pathway/process mapping, Interviews, and surveys, cost current service.	Baseline data, Service cost, Snr service staff views and evidence that there is a senior sponsor.	Sign off at professional level that the CS is a true representation supported by patient views and feedback
	SR1—Demand—Problem validation and Vision	Needs and gaps analysis to identify a clear quantifiable demand/need/gaps	Demand data (ISD, NPI etc.)—testimonials/endorsement at a senior level (e.g. CEO NHS Board, CMO, Gov Director, Minister, Policy lead). Clear vision.	Sign off by SRO and funding partners

Abbreviations: QoL: Quality of Life, QALY: Quality of Adjusted Life Years, NZ: Net Zero, HTA: Heath Technology Assessment, EQIA: Equality Impact assessment, PROMS: Patient reported outcome measures, PREMS: Patient reported experience measures, HRQoL: Health related QoL, RCT: randomised Control Trial, CBA: Cost benefit analysis, SLA: Service Level Agreement, CSF: Critical success factors, BAU: Business as Usual, ROI: Return on Investment, ISD: Information Services, NPI: National performance indicators, CEO: Chief executive officer, CMO: Chief medical officer, SG: Scottish Government, SRO: Senior responsible officer, RWE: real-world evidence.

**Table 3 ijerph-18-12575-t003:** Evidence themes as per Service readiness levels (initial combination framework).

Evidence Themes	Service Readiness Levels (SRL) Framework—Definitions								
	1	2	3	4	5	6	7	8	9
	Demand, Needs and Vision (Assessed and Validated)	Current State (CS) (Agreed and Validated)	Horizon Scanning Landscape Review	Future State (FS) (Co-design) Option Appraisal	Future State (Preferred and Validated)	Real World Evidence (RWE) Testing	Evaluation of the Pilot RWE Site (s)—Evidence Gathered	Case for Scale/(Business Case/ Proposal dev)	Implement at Scale and Improve DHI (as Required)
Service and Organisational evidence	Expert opinion/view/Operational Service stats—local/national (ISD). Vision statement	CS service journey map—Baseline	Publications Case studies and learning/Best practise identified	Service option appraisal (FS maps)/Service support	Preferred Future state service map and path/Comms plan and PR	Change mgmt. review/training/ IG/DSP/BRP/DPIA TP/Risks/EQIA/ Comms pack	SQ/Legal/ HR/IT/IG/IP/ Budget/Risks/ Process eval/ Org outcomes	Blueprint/Imp/ Setup/Impact plan/TM/CM/EQIA/DPIA Comms/Data plan	Strategic/Financial Commercia/Mgmt case/BAU plan—Set up pack—Imp plan/Net Zero action plan
Clinical evidence	Universal view/Baseline demand/SPARRA–Info services/Hypothesis/ Endorsement	CS service journey map—Baseline	Publications/Case studies/Patient safety	FS maps/Multi-disciplinary testimonials/ Efficacy/Ethics	Endorsement Simulation/Leadership/ Ethics/Risk review	Leadership /Change mgmt.—workflow review RCT/marketing	Safety—CRM/CSC /Acceptability/ Effectiveness/ Patient outcomes	CLP/TM/CSA/ Adhérence/ Patient impacts /PR—marketing	Strategic case—Impact/Gov/SOP /Improvement backlog/Comms
Finance, Legal and standards evidence	Approx. costs of demand focus—local/national and Legal/standards view	CS approx. costings—initial costs gathered (if possible)	Cost studies/ Procurement review/Total Cost Factor	Approx. costs—all options/ consequence—‘do nothing’	Cost comparison (CS vs FS)—option review. Net zero considered	Procurement approach view/Ethics app/Economic evaluation	CCA/CEA/TCO/CBA/Affordability and Value for money review. Net Zero contribution plan	CBA/ROI/HTA/CUA/CA/CSv/ Procurment and sustainability. Net zero impact plan	Economic/Comm case/ROI/CBA/GVA NPV + Financial budgets/Net zero impact defined
Citizen evidence	Test citizen views on hypothesis/ Target Population nrs. /Future Demand Projections	CS citizen journey map—baselined/QoL/QALY benchmarking if possible	Publications/Case studies/Best Practise identified/Personas built.	Interview data -view point /PersonasFS map/general requirements	Testimonials on FS appetite /EQIA drafted/risk review. Personas revised	Acceptance/ Accessible/ Usable/Cost to citizen/PROMS/ PREMS/Surveys	UA/UX—Usability data/CtA/QALY QoL/HRQoL PROMS/PREMS/ Survey/Interviews	EQIA/Privacy/Case studies/Benefits & Impacts/HRQoL/ User stories and personas illustrated	Strategic case/Comms and marketing campaign/Training
Political/Policy evidence	Test Political support/Strategic alignment/Policy benefits	Policy/strategic review and priority alignment (Targets + timelines identified) macro costs—system	Political/Priority/importance/ Critical success factors/strategy review (national)	Political support/and sponsorship review—benefit plan—NZ incl.	Endorsement/ Risk review/ /Sponsor local + national level. Benefit plan	Sponsorship /Policy instrument review. Benefit checkpoint	Confirm Sponsorship/ Benefits/NPI/Net zero/EQIA/case outline	Confirm Political/Strategic buy in/CSF/EQIA—quantify social—NZ—economic/benefits	Strategic case Briefing/policy paper/Proposal/ Benefits plan/NZ contribution/Imp Plan
Technical evidence	Tech pull or push—acceptability (consumer demands and appetite to use digital for the focused target groups–popn.)	Existing version of tech/integration/interoperability check and high-level roadmap—baseline	Publications/ Case studies/ref sites.Adaptation/ interoperability review	Tech appraisal /FS alpha dev/Infrastructure/ Integration/UA/UX/PT testing/	HTA/FS Tech architecture map/IMTO/ Simulation and alpha prototype	Data models /Hardware/Software/UI testing Accessibility/beta dev	SSP/IG/PECR/ CE/MDR/FDA/ UA.UX/PT-Pen Tests/IP/W3C WAI/UAT/UX	Business model-costs/TCO User Numbers/HTA/ UAT/Integration plan and costs	Commercial/ Financial case/Sales Comms and PR plan. Service contract and maintenance SLA.

Abbreviations: QoL: Quality of Life, QALY: Quality of Adjusted Life Years, CS: Current state, FS: Future state, NZ: Net Zero, UA: User Acceptance, UX: User Experience, HTA: Heath Technology Assessment, IMTO: Innovative Medical Technology Review, IG: Information Governance, DSP: Data sharing plan, BRP: Benefits realisation plan, DPAI: Data Privacy Impact assessment, TP: training Plan, EQIA: Equality Impact assessment, PROMS: Patient reported outcome measures, PREMS: Patient reported experience measures, SQ: service quality, HR: Human resource, IT: information technology, IP: Intellectual property, HRQoL: Health related Quality of Life, CE: ’Conformité Européene’—European Conformity, FDA: Food and Drugs Administration, MDR: Medical device regulations, CRM: Clinical risk assessment, CEA: Cost effective analysis, CUA: Cost utility analysis, TCO: Total cost of ownership, CBA: Cost benefit analysis, CtA: Contribution analysis, SOP: Standard operating procedures, SSP: System security plan, SLA: Service Level Agreement, UI: User Interface TM: Training manual, CM: Change mgmt. plan, CLP: Clinical protocols, CSA: Clinical safety assurance, CSv: Cost savings, CSF: Critical success factors, BAU: Business as Usual.

## Data Availability

The interview transcripts from this study are embargoed at time of publication due to the confidential nature of some of the content (projects and people named) and ongoing additional analysis and publication of thesis. In addition, the identity of the respondent would be revealed if any large sections of the transcript were available. The processed data (i.e.) Table 2 and Table 3 are provided in the paper.

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
