# Peer review of "Scaling Digital Health Innovation: Developing a New ‘Service Readiness Level’ Framework of Evidence"

_ijerph, 2021, doi:10.3390/ijerph182312575_

Round 1

Reviewer 1 Report

  • The manuscript is written clearly, and it presents a clear picture of the evidence that shows requirements for a project lifecycle to progress a scale. I think the manuscript is acceptable in this current form, however before that I would like to request the authors to address the mentioned comments:
    • I would like to suggest the authors to make a separate section for Related works after introduction section.
    • There should be a flow diagram for framework and also a paragraph to explain it.
    • Findings are difficult to follow. There must be few diagrams and Tables for findings.

Reviewer 2 Report

I think it provides a nice framework.  The details and ordering of information in the paper needs to be further developed.

The manuscript “Scaling Digital Health Innovation: developing a new ‘Service Readiness Level’ framework of evidence” provides results from qualitative interviews of 18 key informants.  The first part of the paper was well written and describes the five themes very well.  However, the discussion and conclusion were less clear, to me.  A few suggestions are provided below.

Given the complexity of this issue, it would be helpful to have more description of the 18 people (age range, sex, level of management, type of institution affiliation…). Always helpful to include year(s) of the data collection as well as duration of the interviews.  If there were semi-structured questions, adding these to the appendix is helpful.  Adding the pre-interview paper would also be helpful.

The discussion was the less connected section.  For example the first sentence states: “What emerged from the interviews and is currently absent from discussion in previous 369

research is an appreciation that the nature of evidence varies dependent on the stage in 370

the lifecycle of the project, criticality of the service, and state of readiness to adopt and 371

scale.”

This was not really well described in the article that listed the themes.  Although it seems common sense, this statement was not really addressed in the paper.

The second sentence refers to the appendix.  ” There was an emerging view that the form of evidence required could be con- 372

ceived of as being linked to different degrees of service readiness levels”

This is odd that the discussion is not about what is written in the text but rather refers to supplemental information.

Both of these statements are critical for the entire concept of adoption and should be more fully incorporated into the text of the article.

Reference to table 2 in the discussion was much more developed and in the text, it seemed to be pointing only to the first column.  Again, the discussion is talking about what has been presented in the method and results section and not new material. 

Overall the discussion needs to be recrafted and material moved from it to the results section,

For the tables, all the acronyms need to be spelled out.  Typically that can efficiently be done as footnote to the able.  Asking readers to go to an appendix is not reader friendly.

In the conclusion:

“validated the potential 438

utility of a heuristic DHI service readiness level framework to detail the type and range 439

of evidence sought to allow DHI scaling to be more easily assessed.”

There was a mention of ‘This SRL framework was then tested and validated by N=14 interviews with key 181

DHI leaders to gain feedback and detail that further optimized the usefulness of the 182

framework.”  But there was little data to support this statement.  The statement ” This testing showed

Table 2 was con- 392

sidered a novel and potentially very ‘useful’ tool in assisting DHI project teams to check 393

at each stage that they were collecting the right evidence and ensuring that the correct 394

content and data was in place for building a robust case for scale.” Is not really formal validation so the statement of validation seems abit strong.  It also was not entirely clear how Table 2 was generated (from the researchers?  From the key informants?) 

How was this validated?  I did not see any data on that? 

“This offers for the 440

first-time clear pathways in how evidence can be accumulated to enable scaling up of 441

digital health innovation and building a robust ‘case for scale’ for decision makers.”

I missed these clear pathways piece and therefore this needs to be more clearly articulated.  Mostly I understood the themes that were generated by 18 people. 

Round 2

Reviewer 1 Report

Authors have adequately addressed the comments and significantly improved the manuscript. I don't have any further comments.